# Multi-Omics Reveals the Role of Arachidonic Acid Metabolism in the Gut–Follicle Axis for the Antral Follicular Development of Holstein Cows

**DOI:** 10.3390/ijms25179521

**Published:** 2024-09-01

**Authors:** Yajun Guo, Shiwei Wang, Xuan Wu, Rong Zhao, Siyu Chang, Chen Ma, Shuang Song, Shenming Zeng

**Affiliations:** State Key Laboratory of Animal Biotech Breeding, National Engineering Laboratory for Animal Breeding, Key Laboratory of Animal Genetics, Breeding and Reproduction of the Ministry of Agriculture, College of Animal Science and Technology, China Agricultural University, Beijing 100193, Chinashiwei@cau.edu.cn (S.W.);

**Keywords:** follicular development, gut microbiota, arachidonic acid metabolism, phosphocholine, steroidogenesis

## Abstract

In vitro embryonic technology is crucial for improving farm animal reproduction but is hampered by the poor quality of oocytes and insufficient development potential. This study investigated the relationships among changes in the gut microbiota and metabolism, serum features, and the follicular fluid metabolome atlas. Correlation network maps were constructed to reveal how the metabolites affect follicular development by regulating gene expression in granulosa cells. The superovulation synchronization results showed that the number of follicle diameters from 4 to 8 mm, qualified oocyte number, cleavage, and blastocyst rates were improved in the dairy heifers (DH) compared with the non-lactating multiparous dairy cows (NDC) groups. The gut microbiota was decreased in *Rikenellaceae_RC9_gut_group*, *Alistipes*, and *Bifidobacterium*, but increased in *Firmicutes*, *Cyanobacteria*, *Fibrobacterota*, *Desulfobacterota*, and *Verrucomicrobiota* in the NDC group, which was highly associated with phospholipid-related metabolites of gut microbiota and serum. Metabolomic profiling of the gut microbiota, serum, and follicular fluid further demonstrated that the co-metabolites were phosphocholine and linoleic acid. Moreover, the expression of genes related to arachidonic acid metabolism in granulosa cells was significantly correlated with phosphocholine and linoleic acid. The results in granulosa cells showed that the levels of PLCB1 and COX2, participating in arachidonic acid metabolism, were increased in the DH group, which improved the concentrations of PGD_2_ and PGF_2α_ in the follicular fluid. Finally, the expression levels of apoptosis-related proteins, cytokines, and steroidogenesis-related genes in granulosa cells and the concentrations of steroid hormones in follicular fluid were determinants of follicular development. According to our results, gut microbiota-related phosphocholine and linoleic acid participate in arachidonic acid metabolism in granulosa cells through the gut–follicle axis, which regulates follicular development. These findings hold promise for enhancing follicular development and optimizing oocyte quality in subfertile dairy cows.

## 1. Introduction

Assisted reproductive technologies have been developed to reduce infertility rates, increase offspring numbers from genetically superior animals, facilitate genetic manipulation, and aid in preservation [1,2]. A potent technique for improving the genetic profile of dairy cows is embryo production, particularly when it comes to transferring genes from females with better genetic profiles and lineages [3,4]. In 2022, the International Embryo Technology Society collected global data on embryo transfer and discovered that the number of embryos produced in vitro in dairy cows surpassed the number of embryos developed in vivo [5,6]. However, owing to lactational metabolic problems, donor cows that are nursing may suffer from reduced oocyte quality, decreased fertilization, and altered embryonic development, leading to decreased fertility [7,8]. The primary reason that cows from cattle farm herds are slaughtered is subfertility; 25% of dairy cows are culled as results of reproduction problems [9]. High milk-output dairy cows have low fertility rates. One contributing aspect is the high blood flow via metabolic pathways in the livers of high-producing dairy cows, which causes the rapid breakdown of estradiol and progesterone [10,11]. Additionally, decreasing circulating progesterone increases the incidence of multiple dominant follicle ovulation and twinning, modulates follicular recruitment in ways that stimulate follicle persistence, and impairs uterine support for embryo and conceptus development [12,13].

As a key component of the oocyte microenvironment, follicular fluid (FF) is vital to oocyte maturation and follicular development [14]. FF-containing active compounds respond to follicular development by influencing granulosa cells’ (GCs) transcription, protein translation, and metabolism, participating in signal communication and substance exchange [14,15]. The ovarian microenvironment facilitates bidirectional communication between GCs and oocytes to maintain normal metabolism of lipids, amino acids, and carbohydrates [16,17]. Metabolite concentrations in FF undergo significant changes during follicular development as consequences of fertility inequality between heifers and lactating cows [13]. Therefore, the quality of oocytes can be assessed and the metabolic conditions of follicles can be reflected through FF metabolite analysis.

The microbiota, together with estrogens, androgens, insulin, and other hormones circulating through the bloodstream, influences the functioning of the reproductive endocrine system, leading to metabolic disorders of reproductive function [18]. Alterations in the gut microbiome are related to several factors, including host genes, hormones, and diet [19,20,21]. Sex hormones are involved in microbe–host communication and exert important roles in reproduction, cell differentiation and proliferation, inflammation, and metabolism [22,23,24,25]. The human microbiome affects female reproduction via follicular development, oocyte maturation, fertilization, embryo migration, implantation, pregnancy, and parturition [18,26]. The reproductive endocrine system is specifically affected by changes in the microbiome, especially the gut microbiome, and improving aberrant microbiomes enhances reproductive outcomes [27,28,29,30]. By modifying amino acid content, the gut microbiota may play a role in the regulation of insulin resistance and lipolysis [31]. Recent studies have shown that modifications in the bacterial composition and metabolism of the gut microbiota can play an indispensable role in the pathogenesis of metabolic process disorders [32,33].

However, the molecular mechanism by which the gut microbiota affects follicular development remains unclear. Hence, the relationship among changes in gut microbiota and metabolism, features of the serum, and FF metabolome atlas will be investigated in the dairy heifers (DH) and non-lactating multiparous dairy cows (NDC) with differences in follicular development after superovulation treatment. Correlation networks will be constructed to decipher how the metabolites affect follicular development by regulating gene expression in GCs. This knowledge provides valuable insights into the gut–follicle axis in dairy cows.

## 2. Results

### 2.1. Comparison of the Efficiency of In Vitro Embryo Production between the DH and NDC Groups

Before exogenous follicle-stimulating hormone stimulation, the DH group showed a relative increase in blood biochemical indices compared with the NDC group, indicating a normal metabolic function of the body (Figure 1A–C). On D0 of OPU, the LH, E_2_, and P_4_ levels were significantly higher in the NDC group than in the DH group (Figure 1C), and on D5, the FSH, LH, E_2_, and P_4_ levels also increased (Figure 1D). After the OPU-IVF, the results showed that the number of antral follicles (30.25 ± 9.70 vs. 21.13 ± 5.93, *p* < 0.05), number of COCs recovered (20.08 ± 9.14 vs. 12.13 ± 7.47, *p* < 0.05), number of qualified oocytes (16.25 ± 7.13 vs. 9.63 ± 5.33), cleavage (72.12 ± 0.07% vs. 41.45 ± 0.25%, *p* < 0.05) and blastocyst rates (28.36 ± 4.43% vs. 16.10 ± 6.60%, *p* < 0.05) were significantly higher in the DH than in the NDC cows (Figure 1F, Appendix A). In general, the NDC group presented with a decreased number of follicles and the poor developmental potential of oocytes during OPU-IVF.

### 2.2. The Difference in the Gut Microbiota Abundance between the DH and NDC Groups

To investigate the differences in gut microbiota abundance between the DH and NDC groups, fecal samples from the two groups were collected for *16S rRNA*. The number of taxonomic units was compared between the microbiomes at different taxonomic levels in the two groups (Figure 2A). We further compared the differences in gut microbiota at the phylum and genus levels between the DH and NDC groups. They were significantly changed at phyla level for *Firmicutes* (58.38% vs. 62.99%), *Bacteroidetes* (34.29% vs. 33.94%), and *Actinobacteriota* (4.56% vs. 0.20%) in the two groups (Figure 2B). The genus level between the DH and NDC groups were *UCG-005* (19.59% vs. 16.92%), *Rikenellaceae_RC9_gut_group* (12.88% vs. 10.90%), *UCG-010* (5.41% vs. 6.30%), *[Eubacterium]_coprostanoligenes_group* (3.94% vs. 5.12%), *Alistipes* (5.09% vs. 3.84%), and *Bacteroides* (4.54% vs. 3.84%) (Figure 2C). The two groups showed no significant differences in Chao1. The α-diversity assessed by the Shannon and Simpson indices exhibited a significant difference between the two groups (Figure 2D). The NDC group had higher species richness than the DH group by the rarefaction curve results (Figure 2E). The overall *β*-diversity of the gut microbiota composition was distinct between the DH and NDC groups (Figure 2F). The results of the unweighted pair-group method with arithmetic means showed that analysis of the abundances of the top 10 phyla and genera in the 2 groups further validated the PCoA results (Figure 2G,H). PCA of the relative abundance of the gut microbial populations indicated that they were separated into two groups (Figure 3A). The Venn diagram of ASV/OTU results showed that there were 2646 differential microorganisms in the 2 groups (Appendix A). The LEfSe results indicated that the *Actinobacteriota* (LDA score = 4.318, *p* = 0.0007) and *Firmicutes* (LDA score = 4.370, *p* = 0.0118) were the discriminatory taxa in the DH and NDC groups (Figure 3B,C). Furthermore, the heatmaps showed the differential expression of 20 bacterial genes and important markers in the 2 groups at the phylum and genus levels (Figure 3D–G). The relative abundance of *Firmicutes*, *Cyanobacteria*, *Fibrobacterota*, *Desulfobacterota*, and *Verrucomicrobiota* was significantly higher in the NDC group than in the DH group (Figure 3D). The relative abundance of *Rikenellaceae_RC9_gut_group*, *Alistipes*, and *Bifidobacterium* was significantly higher in the DH group than in the NDC group (Figure 3F, Appendix A). Functional analysis of the metagenomes based on KEGG pathways showed that functions related to amino acid metabolism, energy metabolism, glycan biosynthesis and metabolism, and lipid metabolism were significantly different between the DH and NDC groups (Appendix A). Correlation analysis showed that *Proteobacteria* were positively correlated with age, weight, body condition, parity, milk yield, FSH, and LH. *Actinobacteriota* and *Spirochaetota* were positively correlated with GLU. The *Rikenellaceae_RC9_gut_group* was positively correlated with cholesterol and HDL levels. *The abundance of Desulfobacteria was* positively correlated with FSH and LH levels (Appendix A). Collectively, these results demonstrate that the gut microbiota may play a critical role in follicular development.

### 2.3. Correlation Analysis Reveals the Relationship between the Gut Microbiota and Its Metabolites

We performed metabolomic profiling of the feces of the two groups. There were 175 differentially expressed metabolites in the DH and NDC groups (Figure 4A,B). Compared to the DH group, we analyzed 26 important differential metabolites and found in the NDC group, the expression levels of 21 metabolites (VIP > 4, *p* < 0.05) were downregulated (Appendix A). Furthermore, differential metabolites were analyzed using the KEGG pathway database to identify the top 10 metabolic pathways (Figure 4C). The heatmap exhibited the relative levels of the differential metabolites of steroid hormone biosynthesis and ABC transporters (Figure 4D,E). Correlation analysis showed that *NK4A214_group*, *Gastranaerophilales*, and *Ruminococcus* were negatively correlated with linoleic acid (*p* < 0.05, Figure 4F). *Bifidobacterium*, *Paeniclostridium*, *Turicibacter*, *Lachnospiraceae_UCG-010*, and *Gastranaerophilales* were positively correlated with 1-stearoyl-sn-glycerol-3-phosphocholine (Figure 4F). Additionally, the correlation network map showed that *Bifidobacterium* was positively correlated with 1-stearoyl-2-arachidonyl-sn-glycero-3-phosphocholine and 1-stearoyl-sn-glycerol-3-phosphocholine (r = 0.8, *p* < 0.01; Figure 4F). These results suggest that gut microbiota-related phosphocholine and linoleic acid may play a critical role in mediating the interaction between the gut microbiota and metabolism.

### 2.4. Correlation Analysis Reveals the Relationship between Serum Metabolic Profiles and Gut Microbes

After the rigorous quality screening of metabolites using the Human Metabolome Database (HMDB), 470 metabolites were annotated and the top 4 most abundant superclasses were lipids and lipid-like molecules (32.77%), organic acids and derivatives (23.38%), organic heterocyclic compounds (9.67%), and benzenoids (7.61%) (Appendix A). The heatmap identified 71 differential metabolites in the 2 groups (Figure 5A,B). The most abundant compounds were 1-hexadecyl-sn-glycero-3-phosphocholine, lysophosphatidylcholine 18:2, 1-pentadecanoyl-sn-glycero-3-phosphocholine, and 1-hexadecanoyl-2-octadecadienoyl-sn-glycero-3-phosphocholine, which were significantly increased in the DH group (VIP > 4, *p* < 0.05; Appendix A). The most significantly enriched pathways were ABC transporters, taurine and hypotaurine metabolism, arginine biosynthesis, biosynthesis of amino acids, lysosomes, and glutathione metabolism (Figure 5C). Correlation analysis revealed a significant correlation between phosphocholine-related metabolites and *Bifidobacterium*, *NK4A214_group*, *UCG-009*, *UCG-002*, and *Gastranaerophilales* (Figure 5D). The correlation network map also showed that *Gastranaerophilales* and *NK4A214_group* were correlated with 1-hexadecyl-2-(8z,11z,14z-eicosatrienoyl)-sn-glycero-3-phosphocholine, 1-(1z-octadecyl)-sn-glycero-3-phosphocholine, and 1-palmitoyl-2-docosahexaenoyl-sn-glycero-3-phosphocholine (Figure 5D). These findings reveal that the phosphatidylcholine in the host is closely associated with gut microbes, which may provide important clues for improving follicular development.

### 2.5. Metabolomic Analysis of FF in the DH and NDC Groups

We identified 899 metabolites using untargeted metabolomic technologies. The top four most abundant superclasses were lipids and lipid-like molecules (36.81%), organic acids and derivatives (21.13%), organoheterocyclic compounds (8.79%), and benzenoids (7.12%) (Appendix A). There were 142 differential metabolites (VIP > 1, *p* < 0.05) in the DH group compared to the NDC group, of which 117 were upregulated and 30 were downregulated (Figure 6A,B). The most abundant metabolites were glycerophospholipids and fatty acids (linoleic acid, 1-oleoyl-sn-glycero-3-phosphocholine, arachidonic acid (peroxide-free), and 1-(1z-hexadecyl)-sn-glycero-3-phosphocholine), the expression of which was significantly upregulated in the DH group (VIP > 4, *p* < 0.05, Appendix A). Furthermore, we analyzed the top pathways mainly enriched in these metabolites by KEGG, including protein digestion and absorption, primary bile acid biosynthesis, ABC transporters, cholesterol metabolism, biosynthesis of amino acids, and mineral absorption (Figure 6C–G). These findings indicate that linoleic acid, arachidonic acid, and phosphocholine levels are associated with follicular development.

### 2.6. Transcriptome Profiles Reveal Functional Gene Expression in GCs

RNA-seq analysis was used to determine the differential expression of genes (DEGs) level of GCs in the DH and NDC groups. The PCA showed that the samples in each group were clustered, suggesting a higher degree of similarity between the samples (Figure 7A). The DEGs analysis results showed that the expression of 2368 genes was upregulated, whereas that of 2442 genes was downregulated in the NDC vs. DH groups (Figure 7B,C). Among them, 211 DEGs were specific to the DH group, whereas 63 genes were expressed in the NDC group. The top 20 GO keywords and 20 pathways were found in each of the 2 groups using GO and KEGG analyses to better understand the roles of these DEGs (Figure 7D–F). Notably, the cell cycle, *p53* signaling pathway, lipids and atherosclerosis, and *Wnt* signaling pathway were significantly enriched in the NDC group compared to the DH group, while oxidative phosphorylation and steroid hormone biosynthesis were downregulated. Moreover, the expression of key marker genes caspase3 (*CASP3*), interleukin 6 (*IL6*), tumor necrosis factor (*TNF)*, and NLR family pyrin domain containing (*NLRP3*) in programmed cell death, were found to be highly expressed in the NDC group compared with the DH group (Figure 7G). In the arachidonic acid pathway, the mRNA levels of arachidonate 12-lipoxygenase (*ALOX12E*) and arachidonate 5-lipoxygenase activating protein (*ALOX5AP*) were higher in the NDC group than in the DH group (Figure 7H). The mRNA levels of cytochrome P450 family 11 subfamily A member 1 (*CYP11A1*), cytochrome P450 family 19 subfamily A member 1 (*CYP19A1*), hydroxy-delta-5-steroid dehydrogenase, 3 beta-and steroid delta-isomerase 1 (*HSD3B1*), and hydroxysteroid 17-beta dehydrogenase 2 (*HSD17B2*) of steroid hormone synthesis were higher in the DH group than in the NDC group (Figure 7I). The mRNA levels of *IGF2BP2*, *IGF2*, *TGFA*, *TGFB1/2*, and *FGF13* of growth factor pathway were highly expressed in the DH compared with the NDC groups (Figure 7J). Taken together, these results indicated that the expression of genes associated with GCs growth, proliferation, and function is essential for follicular development.

### 2.7. Comprehensive Analysis of Different Metabolites in Gut Microbiota, Serum, and FF, and Differentially Expressed Genes

We analyzed the differential metabolites in the two groups to clarify the relationship between metabolites in fecal, serum, and FF in the two groups (Figure 8A–C). Phosphocholine and linoleic acid were found to be co-metabolites in the gut microbiota, serum, and FF (Figure 8D–F). We also found that phosphocholine and linoleic acid were more significantly increased in the gut, serum, and FF in the DH group compared with the NDC group (Figure 8D–F, Appendix A). Furthermore, the relationship between the metabolites and genes was constructed using association analysis, which showed that phosphocholine, linoleic acid, and arachidonic acid were correlated with numerous genes (Appendix A). We performed a co-analysis of the identified linoleic acid and arachidonic acid with genes, which showed that linoleic acid was positively associated with phospholipase A (*PLA2G16*, *PLA2G2A*, and *PLA2G6*), whereas arachidonic acid was positively associated with prostaglandin-endoperoxide synthase 2/cyclooxygenase 2 (*PTGS2/COX2*), *PLA2G4A*, glutathione peroxidase 8 (*GPX8*), and gamma-glutamyl transferase 5 (*GGT5*) (Figure 8G). Collectively, these findings imply that arachidonic acid metabolism is involved in follicular development.

### 2.8. PLCB1/PTGS2 Axis Promotes Follicular Development via COX/PGD_2_ Pathway Dependent on Arachidonic Acid Metabolism

We further investigated how GCs respond to changes in metabolites in the arachidonic acid metabolism and steroid hormone synthesis. The targeted metabolomics results showed that the DH group had significantly higher levels of prostaglandin D_2_ (PGD_2_), prostaglandin F_2α_ (PGF_2α_), arachidonic acid, and docosahexaenoic acid in the arachidonic acid metabolism pathway than the NDC group. However, there were increased levels of thromboxane B2 and leukotriene B4 were significantly higher in the NDC group compared with the DH group (Figure 9A). Notably, the levels of PTGS2 and phospholipase C Beta 1 (PLCB1) were increased in the DH compared with the NDC groups, while arachidonate 5-lipoxygenase (ALOX5) was significantly decreased (Figure 9B,C). Meanwhile, there was an increase in the mRNA expression levels of *CYP19A1*, *CYP11A1*, *CYP17A1*, *HSD17B2*, and *HSD3B1* in the DH group compared with the NDC group. The expression levels of *ALOX5*, *PLCB1*, *IL6*, and *CASP3* were increased in the NDC group compared with the DH group, while the levels of *PTGS2* and migration inhibitory factor (*MIF*) expression were decreased (Figure 9D). In addition, there were increased levels of cholesterol, desmosterol, progesterone, 25-hydroxycholesterol, pregnenolone, *17a*-hydroxyprogesterone, β-sitosterol, and 11-dehydrocorticosteron in the DH group compared to the NDC group, while the levels of 17β-estradiol, androstenedione, testosterone, 17α-hydroxy pregnenolone, and 24-hydroxycholesterol were decreased (Figure 9E,F). Interestingly, there was a strong correlation between steroid hormones in FF and gut microbiota (Appendix A). The correlation analysis further revealed a significant relationship between metabolites in the arachidonic acid metabolism and metabolites of steroid hormones synthesis (Figure 9G). Additionally, the protein expression levels of apoptosis (CASP3, BCL2 associated X (BAX), p38 mitogen-activated protein kinases (p38MAPK), interferon regulatory factor 1 (IRF1), gasdermin-D (GSDMD), and tumor protein p53 (p53) were decreased in the DH compared with NDC groups, while anti-apoptosis factor (MIF, BCL2, superoxide dismutase 1 (SOD1), hypoxia-inducible factor 1 alpha (HIF-1α), and extracellular signal-regulated kinase (ERK1/2)) were increased (Figure 9H–K). Moreover, immunofluorescence results showed a significant positive distribution of COX2 and PLCB1 in the GCs layer in the healthy follicles compared with the atretic follicles, while ALOX5 showed the reverse distribution pattern (Figure 9L). Collectively, the PLCB1/COX2 axis is involved in arachidonic acid metabolism to produce PGD_2_ and PGF_2α_, thereby inhibiting apoptosis and promoting steroid hormone synthesis, which are essential for the function involved in follicular development.

## 3. Discussion

Our results showed that follicle number, qualified oocyte number, cleavage, and blastocyst rates were improved in the DH group after superovulation treatment. Integrating multi-omics results revealed that gut microbiota-related phosphocholine and linoleic acid were involved in the arachidonic acid metabolism by the gut–follicle axis, which played important roles in follicular development. Furthermore, the levels of PLCB1 and COX2, which participate in arachidonic acid metabolism, were elevated in GCs from the DH group, thereby increasing the concentrations of PGD_2_ and PGF_2α_ in FF. The two prostaglandins stimulated the expression of genes associated with survival factors and steroidogenesis in the GCs, while concurrently suppressing the expression of apoptotic genes, thereby fostering follicular development. Unfortunately, we focused only on the essential role of these metabolites in follicular development by influencing the levels of steroid hormones in the follicular fluid and the expression of related proteins in granulosa cells. The specific regulatory mechanisms of how these metabolisms interact correlatively will require in-depth study.

The fertility of lactating dairy cows has experienced a yearly decline of 1% throughout the latter decades of the 20th century and has since persisted at a low level [34]. Aberrant oocyte maturation and reduced oocyte developmental competence can be caused by in vivo and in vitro factors [35,36]. In the present study, the number of follicles and the rates of qualified oocytes, cleavage, and blastocysts decreased in the NDC group. The gut microbiota generates influence beyond the gastrointestinal tract and affects a few physiological events by changing the processes of inflammation and metabolism [23,37]. The environment and host can influence differences in the gut microbiota [38,39]. A diverse gut microbiome, mostly consisting of *Actinobacteria*, *Proteobacteria*, *Firmicutes*, and *Bacteriodetes*, sustained the integrity of the gut epithelial barrier [25,38]. Our results revealed that *Firmicutes* were enriched in the NDC group, whereas *Actinobacteria* and *Bacteroidetes* were enriched in the DH group. Over 90% of the species in a healthy gut microbiome belong to the phyla *Firmicutes* and *Bacteroidetes*; this ratio has been proposed as a diagnostic for microbiota dysbiosis, which is in line with our findings [40,41,42]. Metabolic disorders including obesity and diabetes were closely associated with dysbiosis of the gut microbiota [32]. Because substances in breast milk encourage the growth of the *Bifidobacterium* species, it dominate the infancy microbiome in humans [43,44]. The abundance of *Bacteroidetes* and *Actinobacteria* indicates intestinal flora homeostasis [44]. Our findings showed that *Bacteroides*, *Prevotella*, and *Bifidobacterium* were significantly enriched in the DH group, suggesting that these microorganisms benefit from the metabolites of the organism and enhance cell growth. In addition, the further exploration of the molecular mechanisms of how gut microbiota metabolites act on follicular development remains to be investigated. According to a recent study, the fecal metabolome can be used to understand the role of gut microbes and to characterize the interrelationships between host metabolism and gut metabolism [45]. Phosphocholines and linoleic acid are important energy and bioactive factors to support oocyte growth and maturation by constructing plasma membranes as well as regulating the cell cycle, survival, and apoptosis [46,47,48]. Similarly, we found that the relative abundances of *Rikenellaceae_RC9_gut_group*, *Alistipes*, and *Bifidobacterium* were significantly higher in the DH than in the NDC groups, and these microbes are highly associated with phosphocholine and linoleic acid. These findings provide a basic understanding of how gut bacteria interfere with follicular development. In addition, further exploration is needed as to whether the microorganisms and their metabolites may be a precise potential mechanism influencing follicular development.

The level of FSH was elevated in cows with a low antral follicle count (AFC) when compared to those with a high AFC [49,50]. Similarly, our results showed that the FSH concentration was higher in the NDC group than in the DH group. It was noted that E_2_ concentration also increased, which implied disturbance of the negative feedback mechanism leading to endocrine disorder, hence impairing follicular development in the NDC group. Furthermore, several inflammatory disorders, cancer, and cardiovascular biology were closely linked to the arachidonic acid pathway [51,52,53,54]. Cytoplasmic phospholipase A2 releases arachidonic acid from phosphatidylcholine and phosphatidylethanolamine in the cellular membrane via the PLCB1 pathway [55,56,57]. Free arachidonic acid is metabolized to bioactive eicosanoids via the COX pathway, which can metabolize free polyunsaturated fatty acids (PUFAs) to produce prostaglandins/thromboxanes or the leukotriene family of eicosanoids [58,59]. Notably, our results showed that the levels of PGF_2α_, PGD_2_, arachidonic acid, docosahexaenoic acid, linoleic acid, and phosphocholines were higher in the DH group than in the NDC group in FF. Previous studies have found that the levels of arachidonic acid, LysoPC (16:1), LysoPC (20:4), and LysoPC (20:3) are upregulated, but the levels of LysoPC (18:3) and LysoPC (18:1) are downregulated in young patients compared to in older patients with FF [60]. Lysophosphatidylcholine can be used as a biomarker of follicular stages and ovarian sensitivity to exogenous hormones [61,62]. These results imply that they play a key role in follicular development and oocyte maturation; however, the precise mechanisms require further study. Additionally, arachidonic acid levels decrease during oocyte meiotic resumption, and excess arachidonic acid disrupts oocyte maturation, demonstrating that arachidonic acid metabolism inhibits oocyte maturation, which must be downregulated by arachidonic acid concentration to relieve this inhibitory mechanism [63,64]. Linoleic acid is the most abundant PUFAs in bovine follicular fluid, and its concentration of linoleic acid has previously been reported to decrease significantly with increasing follicular diameters [65]. Supplementation with linoleic acid during bovine oocyte maturation impairs oocyte maturation quality, contributing to a decrease in the percentage of oocytes at stage **metaphase II** as well as the inhibition of subsequent early embryonic development [66]. The above studies and our results suggest that the follicles synthesize prostaglandins/leukotrienes via arachidonic acid/PUFAs catabolism for a possible autocrine function to promote follicular development. Furthermore, it was reported that higher expression of PTGS2 in cumulus cells of the COCs was required for the development of oocytes into high-quality embryos [67]. Other studies have also shown that PTGS2 inhibition during in vitro oocyte maturation leads to abnormal nuclear maturation, with consequent delays in early embryonic development [68,69]. Our study provided evidence that the level of PLCB1 and COX2 in arachidonic acid metabolism was increased in the DH group, which improved the concentrations of prostaglandins in FF. We also found that the levels of apoptosis-related proteins decreased in GCs. Similarly, it has been revealed that PGE_2_ may be a distinct autocrine/paracrine factor, that participates in oocyte maturation and survival [70]. Thus, these results indicate that COX2 accelerates arachidonic acid metabolism to produce PGD_2_ and PGF_2α_, which promote GCs proliferation against apoptosis, implying that the COX2/PGD_2_ pathway could facilitate follicular development.

The follicle is the basic morphophysiological unit of the ovary, as it is the main endocrine and reproductive organ. The GCs express *CYP11A1*, *CYP19A1*, and *17βHSD1*, of which *CYP11A1* is a cleavage of the cholesterol side chain and the primary product is pregnenolone. The primary steroid hormone during the follicular phase, estradiol, is eventually produced by an enzymatic procedure that theca cells utilize to secrete androgens that enter GCs [71,72]. Progesterone, estradiol, and testosterone act to target tissues affecting their activity via the bloodstream to regulate a variety of reproductive events, including apoptosis, inflammation, and metabolism [22]. In our study, the levels of *CYP11A1*, *CYP19A1*, *HSD17B2*, and *HSD3B1* were significantly upregulated in GCs from the DH group. Consistently, the contents of cholesterol, androstenedione, progesterone, pregnanediol, and pregnenolone were increased in FF from the DH cows. Interestingly, we observed a reduction in serum progesterone levels coupled with an increase in progesterone metabolites within the intestinal tract of the DH group. Therefore, we suspected that serum progesterone may traverse the intestinal barrier and enter the gut, potentially modulating the diversity of the gut microbiota, but this hypothesis will need to be confirmed in further research. It was reported that progesterone and estradiol could increase *Prevotella* development [73], and progesterone improved *Bifidobacterium* richness in the gut of women and mice [74]. Consistently, in our results, the abundance of *Bifidobacterium* and *Prevotellanceae* was increased in the DH group. In contrast, sex steroid hormones may regulate gut microbiota to produce bioactive metabolites that influence circulating hormone levels by fine-tuning the balance between their excretion and reuptake. The abundance of *Bifidobacterium* also showed a strong positive correlation with gut microbiota-related phosphocholine in the DH group, which has been proven to regulate steroidogenesis related to gene expression during follicular development. However, the molecular mechanism of the interaction between gut microbes and steroid hormones in dairy cows remains unclear and needs further exploration. Collectively, the present findings suggest that bidirectional communication between steroid hormones and gut microbiota plays a key role in modifying host bacterial action and follicular development.

## 4. Conclusions

Our results suggest that gut microbe-associated phosphocholine and linoleic acid mediate arachidonic acid metabolism through the gut–follicular axis by involving the PLCB1/COX2 pathway in granulosa cells, facilitating the synthesis of PGD_2_ and PGF_2α_, which decreases the levels of apoptosis-associated proteins, facilitates the expression of steroidogenesis-associated genes, and increases the levels of steroid hormones to achieve normal follicular development.

## 5. Materials and Methods

### 5.1. Animals

The DH (*n* = 8) and NDC (*n* = 8) groups in this experiment were provided by the Beijing Dairy Cattle Center. Feeding all dairy cows according to a nutritional formula with the same mixing ratios was aimed at maintaining normal body demands and metabolism. Appendix A provides specific information for each cow. The China Agricultural University Laboratory Animal Welfare and Animal Experimental Ethical Inspections Committee approved the experimental protocol.

### 5.2. Samples Collection and Study Design

Appendix A shows the process and design of this study. On day 0 of ovum pick-up (OPU), clot-activating tubes were used to collect blood from the tail vein. After centrifugation at 500× *g* for 5 min, the serum was stored at −80 °C for the detection of hormones (*n* = 3) and biochemical indicators (*n* = 3). On day 5 of OPU, Serum was obtained from the tail vein for untargeted metabolomics (*n* = 6) and hormone analyses (*n* = 3). Feces were collected from the rectum by hand using gloves, after which samples were collected using a sterilized disposable tool for the detection of 16S rRNA sequencing (*n* = 8) and untargeted metabolomics (*n* = 6). Finally, the cows were anesthetized and OPU was used for transvaginal follicular aspiration. Immediately after aspiration, the mixture obtained included FF, cumulus–oocyte complexes (COCs), and GCs. Specifically, FF was centrifuged at 500× *g* for 5 min, and the supernatant solution was collected and frozen at −80 °C for untargeted metabolomics (*n* = 6) analysis. GCs were washed with PBS by centrifugation at 300× *g* for 5 min and stored at −80 °C for RNA sequencing (*n* = 4). We selected FF and GCs samples from the two groups for targeted metabolomics (*n* = 4), Western blotting, and quantitative real-time PCR analysis.

### 5.3. Superovulation Synchronization and In Vitro Fertilization (IVF)

Appendix A illustrates the treatment protocol for the superovulation synchronization of cows. Briefly, on D0, B-ultrasound (KX5600, Xuzhou, China) was performed and follicles (diameter > 5 mm) were punctured using a device (EXAPad, IMV Imaging, Angoulême, France). The Appendix A details the specific operation method of the OPU. On D5, before OPU, the cows were taken to the lab, securely bound, and administered epidural anesthesia using a 2% lidocaine solution. The COCs in the antral follicles were retrieved using a puncture instrument adapted for bovine ovary aspiration. The dish was rinsed with modified DPBS following aspiration, and the search for COCs was performed using a stereomicroscope. The COCs were matured at 38.5 °C under 5% CO_2_ in humidified air for 24 h. The COCs were incubated with the final spermatozoa concentration (2 million sperm/mL) in a medium for 10 h at 38.5 °C under 5% CO_2_ in a humidified atmosphere. After fertilization, the cumulus cells and sperm surrounding the presumptive zygotes were removed by gentle pipetting and incubated in a medium at 38.5 °C under 5% CO_2_ and 5% O_2_ in a humidified atmosphere until blastocysts were present. All the culture media were obtained from IVF Bioscience (Falmouth, Cornwall, UK).

### 5.4. Fecal 16S rRNA Sequencing and Processing

Whole genomic DNA was extracted from fecal samples. The bacterial 16S rRNA gene V3–V4 region (forward primer 338F (5′-ACTCCTACGGGAGGCAGCA-3′) and reverse primer 806R (5′-GGACTACHVGGGTWTCTAAT-3′)) were amplified by PCR. Following the individual quantification step, amplicons were pooled in equal proportions, and pair-end 2 × 250 bp sequencing was carried out using the Illumina NovaSeq platform and the NovaSeq 6000 SP Reagent Kit (500 cycles). The DADA2 plugin was used for quality filtering, denoising, merging, and chimera elimination from the sequences. Non-singleton amplicon sequence variations (ASVs) were used to build a phylogeny using fasttree2 after they were aligned with a mat. Principal coordinate analysis (PCoA), beta diversity analysis, principal component analysis (PCA), and alpha diversity indices were among the sequence data analyses carried out with the use of QIIME2 (v1.9.1) and R packages (v3.2.0). Using default parameters, differentially abundant taxa across groups were identified using linear discriminant analysis effect size (LEfSe). Phylogenetic analysis of communities by reconstruction of unobserved states (PICRUSt2) allowed the prediction of microbial functions using the Kyoto Encyclopedia of Genes and Genomes (KEGG) and Metacyc (https://metacyc.org/) databases (accessed on 20 December 2023).

### 5.5. Untargeted Metabolomic Analysis for Gut Microbiota, Serum, and FF

FF, blood, and feces were subjected to untargeted metabolomic analysis using liquid chromatography-mass spectrometry (LC-MS). After dissolving the samples in 100 μL of acetonitrile/water solvent, they were centrifuged at 14,000× *g* at 4 °C. The supernatant was examined using quadrupole time-of-flight (AB Sciex Triple TOF 6600, Framingham, MA, USA) in conjunction with ultra-high-performance liquid chromatography (1290 Infinity LC, Agilent Technologies, Santa Clara, CA, USA). A = 0.5 mM ammonium fluoride in water and B = acetonitrile made up the mobile phase in ESI negative mode, whereas in ESI positive mode, the mobile phase comprised A = water with 0.1% formic acid and B = acetonitrile with 0.1% formic acid. XCMS software (v3.12.0, Scripps Research Institute, La Jolla, CA, USA) was used to analyze the raw data. R package ropes were used to analyze the processed data. Multivariate analyses, such as Pareto-scaled PCA and orthogonal partial least-squares discriminant analysis (OPLS-DA), were performed on the data. Statistical significance was determined using the unpaired Student’s *t*-test. In terms of statistical significance, metabolites with variable importance in projection (VIP) values > 1 (*p* < 0.05) were deemed significant. MetaboAnalyst (https://www.metaboanalyst.ca/, accessed on 1 January 2024) was used to perform the KEGG enrichment analysis of several metabolites.

### 5.6. Transcriptomic Analysis of Follicular GCs

Total RNA was extracted from GCs using the TRIzol reagent (Takara, Dalian, China). Libraries for transcriptome sequencing were created using the Illumina Nebnext Ultra Directional RNA Library Prep Kit (New England Biolabs Inc; Ipswich, Massachusetts, USA). An Illumina NovaSeq 6000 sequencer was used for the transcriptome sequencing. Samples were sequenced on the platform to produce picture files and original data in the FASTQ format. The sequencing data were filtered using Cutadapt (v1.15) to provide high-quality sequences for additional examinations. With HISAT2 v2.0.5, the filtered reads were mapped to the reference genome. HTSeq (0.9.1) statistics were used to compare the read count values for each gene with its original expression. Fragments per kilobase of transcript per million fragments mapped (FPKM) were then used to normalize expression. DESeq (1.30.0) was used to evaluate differentially expressed genes (DEGs), with the screening requirements being |log2FoldChange| > 1 and *p* < 0.05. The heatmap was clustered using the full linkage approach and the distance was calculated using the Euclidean method. GO and KEGG analysis were used to examine the DEGs in more detail.

### 5.7. Quantitative Real-Time PCR Analysis

Quantitative real-time PCR was performed using real-time PCR equipment (Bio-Rad, Hercules, CA, USA), HiScript R III-RT SuperMix, and Taq Pro Universal SYBR qPCR Master Mix (Vazyme, Nanjing, China) for first-strand cDNA synthesis and PCR master mix preparation, according to the manufacturer’s instructions. Sangon Biotech (Shanghai, China) produced specific oligonucleotide primers to amplify target genes (Appendix A). *GAPDH* was used as a control to calculate the relative mRNA expression using the 2^−ΔΔCt^ technique.

### 5.8. Western Blotting

RIPA buffer was used for protein extraction (Solarbio, Beijing, China). Protein quantification was performed using a bicinchoninic acid assay (BCA) kit. Protein lysates were electrotransferred by SDS-PAGE onto polyvinylidene difluoride membranes (PVDF) and blocked in TBST containing 5% non-fat milk for 1 h at 37 °C. The PVDF membrane was incubated with the primary antibody at 4 °C overnight, washed with TBST, and incubated with the secondary antibody at 37 °C for 1 h. The ECL Plus Western blotting Detection System (GE Healthcare, Piscataway, NJ, USA) was used to detect the protein bands. Antibody information is presented in Appendix A.

### 5.9. Immunofluorescence Staining

Ovaries from a local cow abattoir were fixed in 4% paraformaldehyde for 48 h and then processed into 5 µm thick tissue sections. The tissue was fixed, dehydrated, wax drained, embedded, and sectioned. Paraffin sections were incubated with the primary antibody at 4 °C overnight and with a secondary antibody (Appendix A) at 37 °C for 1 h. Cell nuclei were stained with 4′,6-diamidino-2′-phenylindole (DAPI) for 5 min at 25 °C and then covered with coverslips. Images were obtained using a laser scanning confocal microscope (LSM 710, Zeiss, Oberkochen, Germany).

### 5.10. Blood Physiological Indices and Hormone Analysis

Blood samples were analyzed using an automatic biochemistry analyzer (TBA-120FR, Toshiba, Tokyo, Japan), including alanine transaminase (ALT), aspartate aminotransferase (AST), glucose (GLU), and cholesterol (CHOL). An XH6080 immunoassay analyzer (Beijing Furuirunze Biotechnology Co., Beijing, China) was used to measure serum hormones including follicle-stimulating hormone (FSH), luteinizing hormone (LH), estrogen (E_2_), cortisol (COR), and progesterone (P_4_).

### 5.11. Targeted Metabolite Analysis

Standard stock solutions (Olchemlm Ltd., Olomouc, Czech Republic) were prepared at a concentration of 1 mg/mL in methanol (MeOH; Merck, Darmstadt, Germany. The FF samples were thawed and vortexed for 10 s, 400 μL of the supernatant was added to the Eppendorf tube and concentrated until completely dry, and 80 μL of the supernatant was transferred for LC-MS analysis. The LC-ESI-MS/MS system (UPLC, ExionLC AD, https://sciex.com.cn/; MS, QTRAP^®^ 6500+ System) was used to analyze the sample extracts (accessed on 14 January 2024). The analytical conditions were as follows: HPLC, 30% acetonitrile/water with 0.04% acetic acid (A) and 50% acetonitrile/isopropanol with 0.04% acetic acid (B); Phenomenex Kinetex C18 (1.7 µm, 100 mm × 2.1 mm i.d.). The gradient was ramped back to 5% B (12.6–15 min) after having climbed to 90% B (1.0–10), 95% B (10–12.5), and 5% B (0–1.0 min). With its ESI Turbo Ion-Spray interface, the AB 6500+ QTRAP^®^ LC-MS/MS System (SCIEX, Framingham, MA, USA) may function in both positive and negative ion modes. It was controlled by Analyst 1.6 software (AB Sciex). Multiquant software (version 3.0.2) was used to obtain the peak areas and retention times of the chromatograms. Appendix A showed the standard curve and retention time for quantification of the chosen target compounds.

### 5.12. Statistical Analysis

Data are presented as the mean ± standard deviation (SD). A two-tailed unpaired Student’s *t*-test was used to calculate *p* values. Version 9 of GraphPad Prism (GraphPad Software, La Jolla, CA, USA) was used for statistical analysis. Statistical significance was set at *p* < 0.05. The figures and figure legends provide statistical parameters.

## Figures and Tables

**Figure 1 ijms-25-09521-f001:**
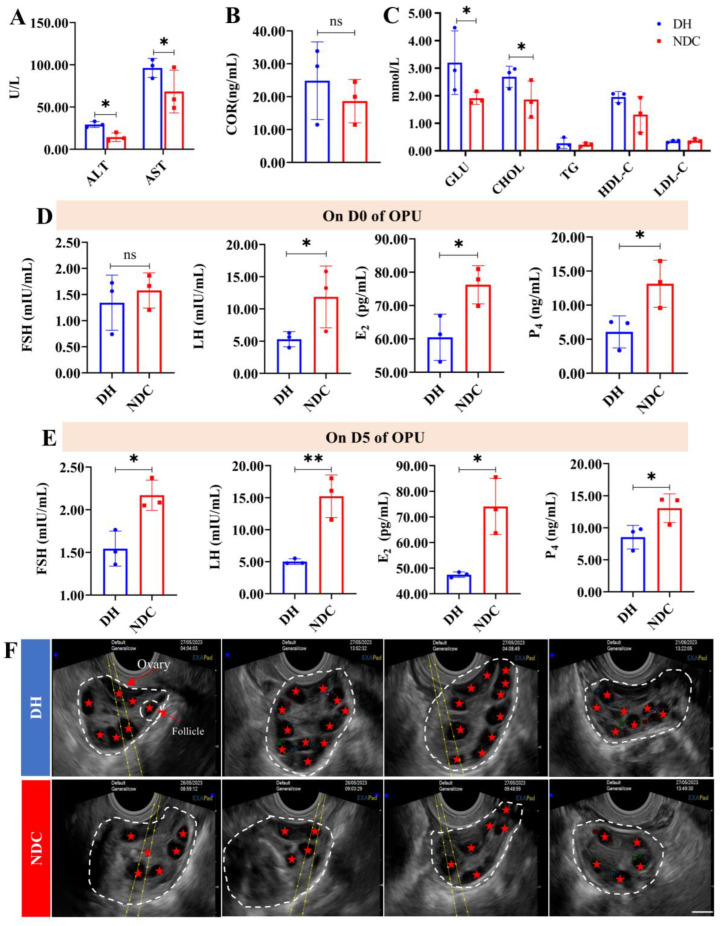
Serum concentrations of key biochemicals and reproductive hormones and antral follicle numbers at OPU in the DH and NDC groups. (**A**) Serum levels of alanine transaminase (ALT) and aspartate aminotransferase (AST) on D0 of OPU in the DH and NDC groups (*n* = 3 per group). (**B**) Serum cortisol (COR) levels on D0 of OPU in the DH and NDC groups (*n* = 3 per group). (**C**) Serum levels of triglycerides (TG), glucose (GLU), cholesterol (CHOL), high-density lipoprotein cholesterol (HDL-C), and low-density lipoprotein cholesterol (LDL-C) on D0 of OPU in the DH and NDC groups (*n* = 3 per group). (**D**) Serum levels of follicle-stimulating hormone (FSH), luteinizing hormone (LH), estrogen (E_2_), and progesterone (P_4_) on D0 of OPU in the DH and NDC groups (*n* = 3 per group). (**E**) Serum levels of FSH, LH, E_2,_ and P_4_ on D5 of OPU in the DH and NDC groups (*n* = 3 per group). (**F**) Ultrasound images of follicular development on D5 of OPU in the DH and NDC groups. The white dashed box indicates the ovary and the red pentagonal star indicates the follicle. Bar scale 8 mm. Data are presented as mean ± SD. Student’s *t*-test (two-tailed) was used for statistical analysis. * *p* < 0.05, ** *p* < 0.01. ns: not significant. DH: dairy heifers, NDC: non-lactating multiparous dairy cows.

**Figure 2 ijms-25-09521-f002:**
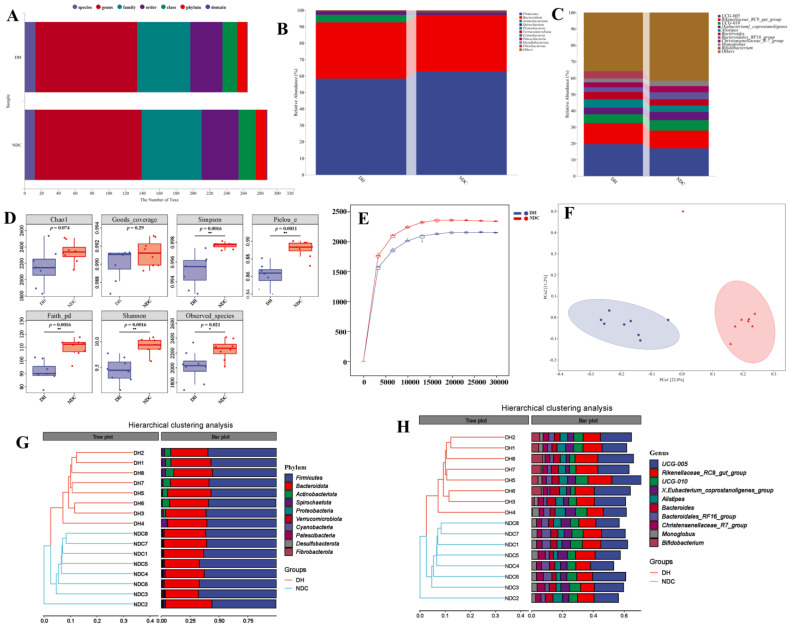
Composition of the gut microbiota in the DH and NDC groups. (**A**) Taxonomic annotation of gut microbiota for DH and NDC samples. (**B**) Relative abundance of gut microbiota at the phylum level in the DH and NDC samples. (**C**) Relative abundance of gut microbiota at the genus level in the DH and NDC samples. (**D**) Alpha diversity evaluation of gut microbiota richness and evenness by measuring Chao and Shannon diversity indexes. *p* < 0.05 significant difference, *p* < 0.01 highly significant difference, *p* ≥ 0.05 insignificant difference. (**E**) Rarefaction curves of gut microbiota for DH and NDC groups. (**F**) Bray-Curtis Principal coordinate analysis plot of gut microbiota based on the operational taxonomic unit metrics of the samples in the DH and NDC groups. (**G**) Unweighted pair-group method with arithmetic mean (UPGMA) analysis at the phylum level for DH and NDC samples. (**H**) UPGMA analysis at the genus level for DH and NDC samples. *n* = 8 per group.

**Figure 3 ijms-25-09521-f003:**
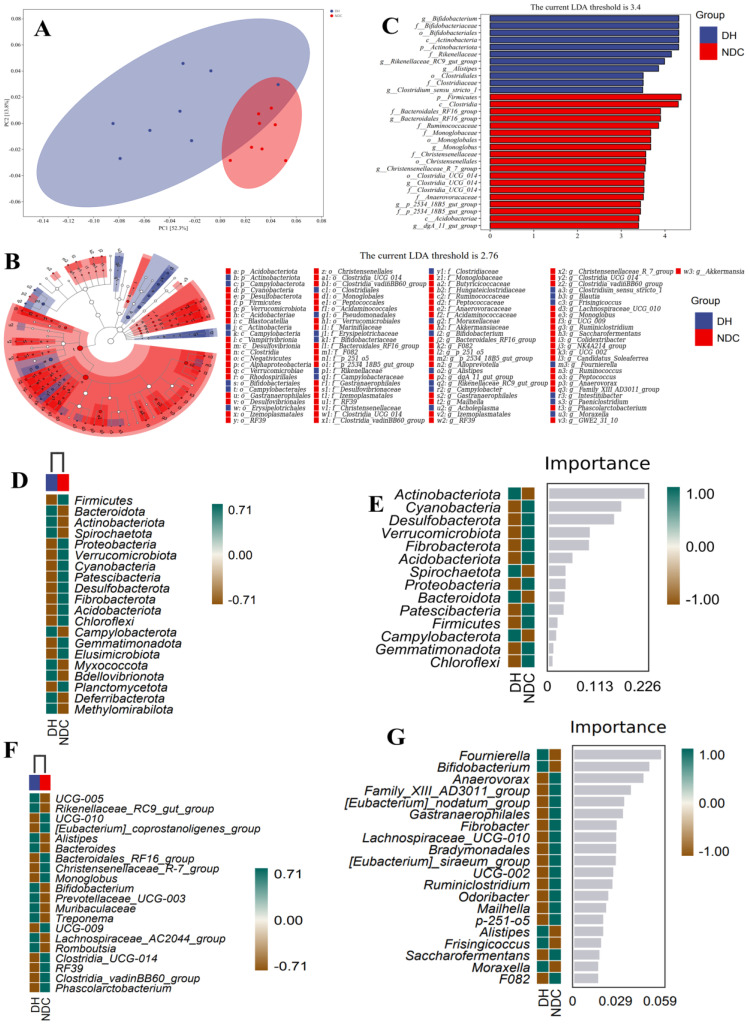
Identification of key differential gut microbiota in the DH and NDC groups. (**A**) Principal coordinates analysis (PCoA) for the DH and NDC groups. (**B**,**C**) Linear discriminant analysis effect size (LEfSe) was performed to identify the differential microbiota in the DH and NDC groups. (**D**) Heatmap showing the differences in gut microbiota abundance at the phylum level between the DH and NDC groups. (**E**) The random forest analysis demonstrated the importance ranking of differential gut microbes at the phylum level between the DH and NDC groups. (**F**) Heatmap showing the difference in gut microbiota abundance at the genus level between the DH and NDC groups. (**G**) The random forest analysis demonstrated the importance ranking of differential gut microbes at the genus level in the two groups. *n* = 8 per group.

**Figure 4 ijms-25-09521-f004:**
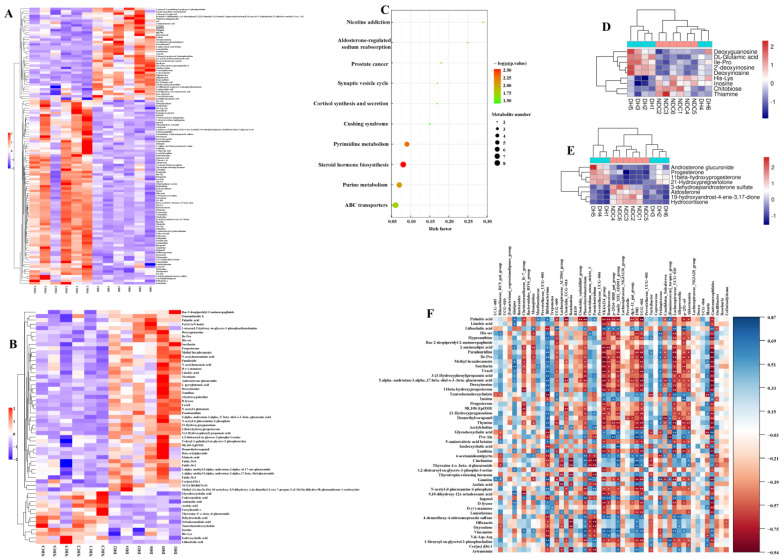
The composition and different metabolites of gut microbiota and correlation with gut microbiota between the DH and NDC groups. (**A**,**B**) Heatmap showing the relative abundance of key identified metabolites (VIP > 1, *p* < 0.05). (**C**) KEGG enrichment analysis of differential metabolites. (**D**) Relative levels of differential metabolites of ABC transporters. (**E**) Relative levels of the differential metabolites of steroid hormone biosynthesis. (**F**) Spearman’s correlation analysis of different gut microbiota and metabolites. Blue indicates a positive correlation and red indicates a negative correlation. * *p* < 0.05, ** *p* < 0.01.

**Figure 5 ijms-25-09521-f005:**
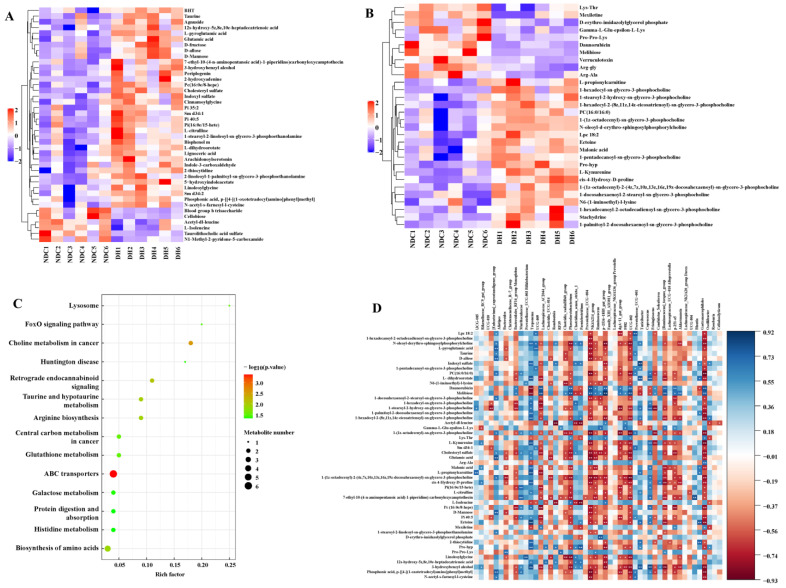
Composition and differences in serum metabolites between the DH and NDC groups. (**A**,**B**) Heatmap showing the relative abundance of the key identified metabolites (VIP >1, *p* < 0.05. *n* = 6 per group). (**C**) KEGG enrichment analysis of differential metabolites. (**D**) Correlation analysis of gut microbiota and serum metabolites. Blue indicates a positive correlation and red indicates a negative correlation. * *p* < 0.05, ** *p* < 0.01. *n* = 6 per group.

**Figure 6 ijms-25-09521-f006:**
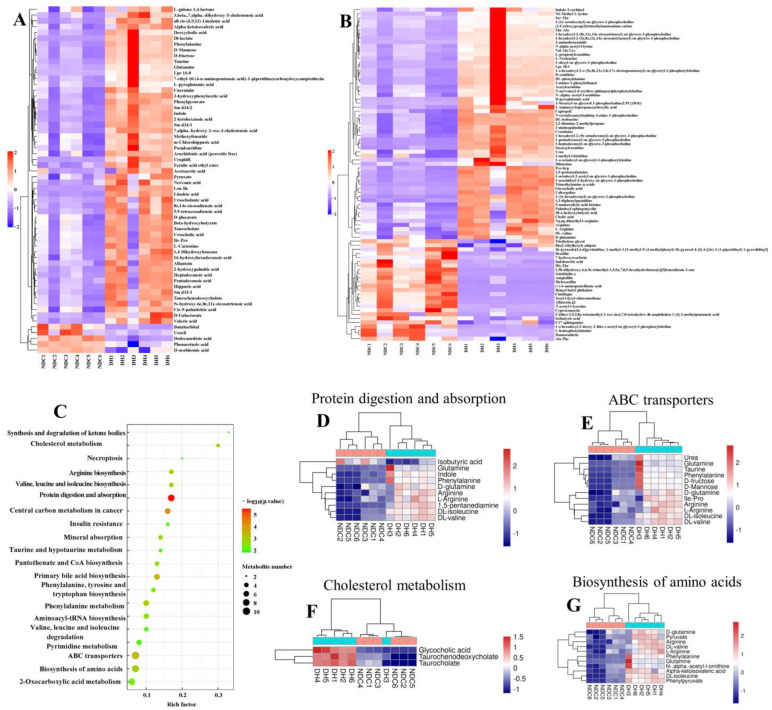
Composition and different metabolites in the follicular fluid between the DH and NDC groups. (**A**,**B**) Heatmap showing the relative abundance of the key identified metabolites (VIP > 1, *p* < 0.05). (**C**) KEGG enrichment analysis of different metabolites. (**D**) Relative levels of different metabolites of protein digestion and absorption. (**E**) Relative levels of different metabolites of ABC transporters. (**F**) Relative levels of the differential metabolites of cholesterol metabolism. (**G**) Relative levels of different biosynthesis of amino acids metabolites. *n* = 6 per group.

**Figure 7 ijms-25-09521-f007:**
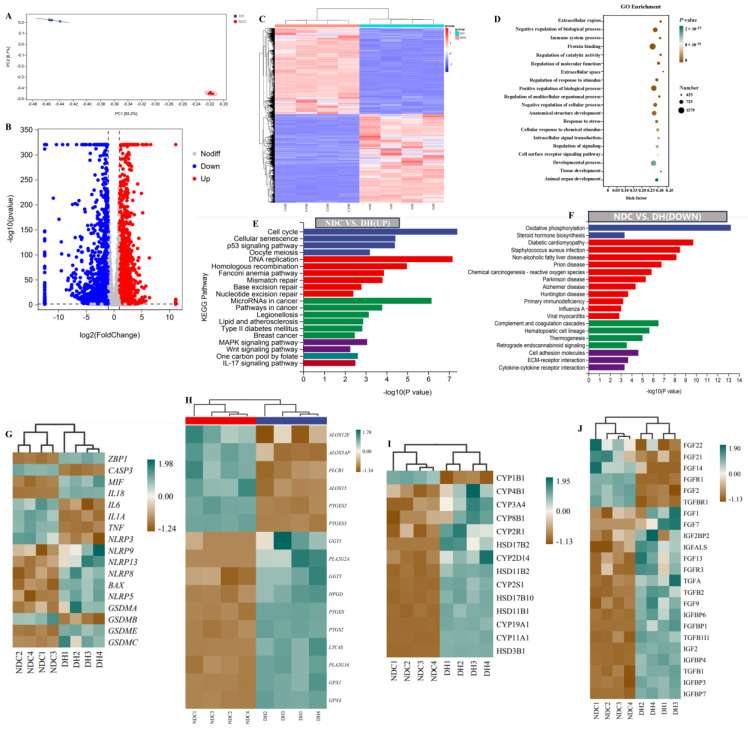
The different expression of genes in granulosa cells between the DH and NDC groups. (**A**) PCA analysis. (**B**) Volcano plot showing the changes in gene expression. (**C**) Heatmap of differentially expressed genes (DEGs). (**D**) GO enrichment analysis of DEGs. (**E**) KEGG pathway enrichment analysis of up-regulation DEGs. (**F**) KEGG pathway enrichment analysis of down-regulation DEGs. (**G**) Heatmap illustrating the expression of genes related to apoptosis and programmed cell death in the two groups. (**H**) Heatmap illustrating the expression of genes related to arachidonic acid metabolism in DH and NDC groups. (**I**) The heatmap illustrates the expression of genes related to steroid hormone metabolism in the two groups. (**J**) Heatmap illustrating the expression of genes related to growth factors in the two groups. *n* = 4 per group.

**Figure 8 ijms-25-09521-f008:**
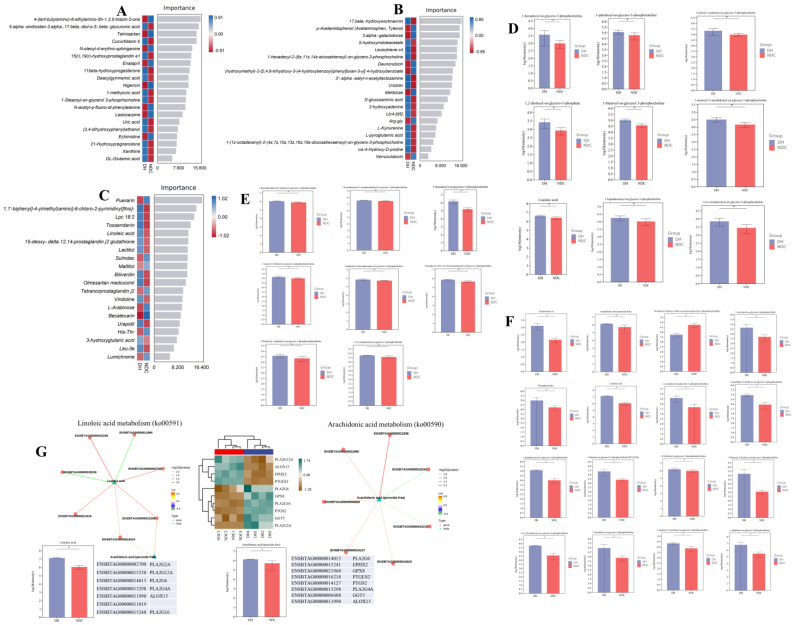
Screening for shared metabolites among gut microbiota, serum, and follicular fluid to construct their regulatory network in granulosa cells. (**A**–**C**) The important metabolites in gut microbiome (**A**), serum (**B**), and follicular fluid (**C**) by the random forest analysis. (**D**–**F**) Expression of metabolites related to membrane phospholipids in rectal feces (**D**), serum (**E**), and follicular fluid (**F**). (**G**) The network maps in linoleic acid and arachidonic acid metabolism pathways. The red line showed a positive correlation and the green line indicates a negative correlation (|R| > 0.8 and *p* < 0.01). * *p* < 0.05.

**Figure 9 ijms-25-09521-f009:**
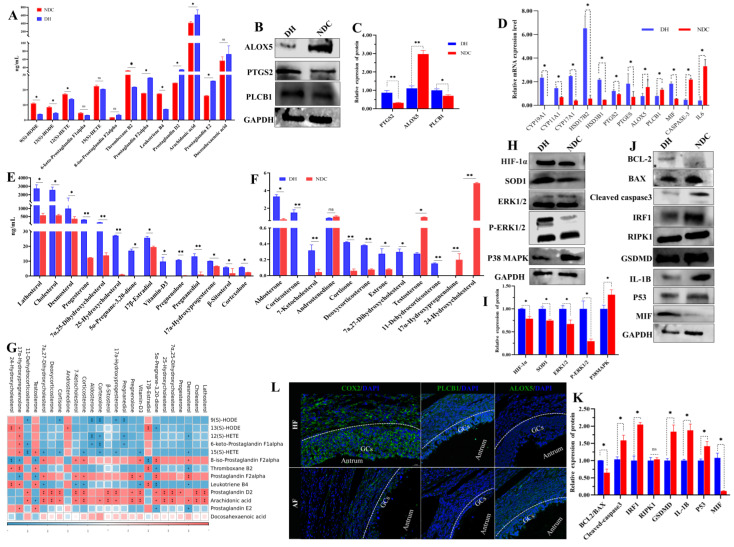
Metabolite levels in follicular fluid related to arachidonic acid and steroid hormones, and gene expression in granulosa cells related to both pathways. (**A**) Levels of arachidonic acid metabolites in follicular fluid (*n* = 3 per group). (**B**,**C**) The levels of PTGS2, ALOX5, and PLCB1 in granulosa cells. (**D**) The mRNA expression levels of genes related to apoptosis, arachidonic acid metabolism, steroid hormones synthesis, and inflammation in granulosa cells. (**E**,**F**) The levels of steroid hormones metabolites in follicular fluid (*n* = 3 per group). (**G**) Correlation between metabolites in the arachidonic acid metabolism and steroid hormones synthesis. (**H**,**I**) The levels of HIF-1α, SOD1, ERK1/2, P-ERK1/2, and p38MAPK in granulosa cells. (**J**,**K**) The levels of apoptosis-related proteins (BCL2, BAX, CASP3, and P53), and programmed cell death-related proteins in granulosa cells (IRF1, RIPK1, GSDMD, IL-1B, and MIF). (**L**) Comparison of COX2, PLCB1, and ALOX5 expression between healthy (HF) and atresia (AF) follicles by immunofluorescence staining. COX2, PLCB1, and ALOX5 were stained green. DNA was stained blue. White dashed line indicates follicular basement membrane. Scale bar, 20μm. Data are presented as mean ± SD. Student’s *t*-test (two-tailed) was used for statistical analysis. * *p* < 0.05, ** *p* < 0.01, ns: not significant.

## Data Availability

All raw sequences were deposited in the NCBI Sequence Read Archive under the accession numbers SRP475896 (microbiota raw sequence data) and SRP475596 (RNA-seq raw data). Further data requests and more information can be obtained by contacting the corresponding authors.

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
