# Peer review of "Multi-Omics Reveals the Role of Arachidonic Acid Metabolism in the Gut–Follicle Axis for the Antral Follicular Development of Holstein Cows"

_ijms, 2024, doi:10.3390/ijms25179521_

Round 1

Reviewer 1 Report

Comments and Suggestions for Authors

The research topic, design of experiments and resources used for such study are highly relevant. I will be glad to continue with the revision of this article after the authors will fix the Figures resolution on this article since it is really difficult to understand the results (quite impossible, since there are a lot of results which need interpretation and correlation between each other) without visual representation. 

Author Response

Point 1: -The research topic, design of experiments and resources used for such study are highly relevant. I will be glad to continue with the revision of this article after the authors will fix the Figures resolution on this article since it is really difficult to understand the results (quite impossible, since there are a lot of results which need interpretation and correlation between each other) without visual representation. 

Response 1: Thanks for your suggestion. Figures have been updated in the manuscript.

Reviewer 2 Report

Comments and Suggestions for Authors

Comments about the manuscript:

“Multi-Omics Reveals the Role of Arachidonic Acid Metabolism in the Gut–Follicle Axis for the Antral Follicular Development of Holstein Cows”

In vitro embryonic technology is hampered by poor oocyte quality and insufficient developmental potential. The aim of the work presented here was to study the relationships that may exist between modifications of the intestinal microbiota, the composition of serum and the metabolome of follicular fluid. Correlations revealed how ovarian follicle development was affected by metabolites involved in the regulation of granulosa gene expression. The particularly numerous and promising results have given rise to figures that are particularly rich in results (but which require improvement in their presentation). They will certainly improve the development of ovarian follicles and optimize the quality of eggs in subfertile dairy cows.

The article is well written, comprehensible to all scientists despite the very large number of results. The spplementary file is useful and does not make the already substantial text heavier.

This important work could be published after some improvements to the manuscript. Here are some remarks.

Abbreviations: there are many abbreviations, so it would be helpful to give a list at the end of the main text or in “materials and methods”.

Page 3, line 103, figure 1: the captions are very small and difficult to read, even after being enlarged by computer.  The figure needs to be improved.

Figure 1 F, ultrasound images: captions or arrows showing important parts of the image would be helpful. A scale bar should be added to the image.

Page 5, figure 2, “Composition of the gut microbiota”: the captions are very small and difficult to read, even after being enlarged by computer. The figure needs to be improved.

Page 6, Figure 3, “Identification of the key differential gut microbiota”: the captions are very small and difficult to read, even after being enlarged by computer. The figure needs to be improved.

Page 7, Figure 4, “The composition and different metabolites of gut microbiota”: the captions are very small and difficult to read, even after being enlarged by computer. The figure needs to be improved.

Page 7, line 208. “After rigorous quality screening and identification”. What was the method used?

Page 8, Figure 5, “The composition and difference of serum metabolites”: the captions are very small and difficult to read, even after being enlarged by computer. The figure needs to be improved.

Page 9, Figure 6, “Composition and different metabolites in the follicular fluid”: the captions are very small and difficult to read, even after being enlarged by computer. The figure needs to be improved.

Page 11, Figure 7, “The different expression of genes in granulosa cells”: the captions are very small and difficult to read, even after being enlarged by computer. The figure needs to be improved.

“figure 7I” is not indicated in the legend.

Page 12, Figure 8, “Screening for shared metabolites among gut microbiota, serum, and follicular fluid”: the captions are very small and difficult to read, even after being enlarged by computer.  he figure needs to be improved.

Page 15, Figure 9, “Metabolite levels in follicular fluid related to arachidonic acid and steroid hormones”: the captions are very small and difficult to read, even after being enlarged by computer. The figure needs to be improved. Scale bars are required for Figure 9K.

“figure 9A” is not indicated in the legend.

Page 19, line 563. “Following the manufacturer's instructions” is not enough for a scientific article: briefly describe the method.

Page 19, line 594. “5.8. Immunofluorescence Staining”. Were negative controls (without primary antibody) performed? Indicate the results obtained.

Author Response

Point 1: - Abbreviations: there are many abbreviations, so it would be helpful to give a list at the end of the main text or in “materials and methods”.

Response 1: Thanks for the helpful suggestion. Abbreviations have been added to the manuscript.

Point 2: -Page 3, line 103, figure 1: the captions are very small and difficult to read, even after being enlarged by computer.  The figure needs to be improved.

Response 1: Thanks for the helpful suggestion. The figure has been updated.

Point 3: -Figure 1 F, ultrasound images: captions or arrows showing important parts of the image would be helpful. A scale bar should be added to the image.

Response 1: Thanks for the helpful suggestion. Scales have been added as required.

Point 4: -Page 5, figure 2, “Composition of the gut microbiota”: the captions are very small and difficult to read, even after being enlarged by computer. The figure needs to be improved.

Response 1: Thanks for the helpful suggestion. The figure has been updated.

Point 5: -Page 6, Figure 3, “Identification of the key differential gut microbiota”: the captions are very small and difficult to read, even after being enlarged by computer. The figure needs to be improved.

Response 1: Thanks for the helpful suggestion. The figure has been updated.

Point 6: -Page 7, Figure 4, “The composition and different metabolites of gut microbiota”: the captions are very small and difficult to read, even after being enlarged by computer. The figure needs to be improved.

Response 1: Thanks for the helpful suggestion. The figure has been updated.

Point 7: -Page 7, line 208. “After rigorous quality screening and identification”. What was the method used?

Response 1: Thanks for the helpful suggestion. Metabolites identification using the Human Metabolome Database (HMDB) databases. Modified in the manuscript.

Point 8: -Page 8, Figure 5, “The composition and difference of serum metabolites”: the captions are very small and difficult to read, even after being enlarged by computer. The figure needs to be improved.

Response 1: Thanks for the helpful suggestion. The figure has been updated.

Point 9: -Page 9, Figure 6, “Composition and different metabolites in the follicular fluid”: the captions are very small and difficult to read, even after being enlarged by computer. The figure needs to be improved.

Response 1: Thanks for the helpful suggestion. The figure has been updated.

Point 10: -Page 11, Figure 7, “The different expression of genes in granulosa cells”: the captions are very small and difficult to read, even after being enlarged by computer. The figure needs to be improved.“figure 7I” is not indicated in the legend.

Response 1: Thanks for the helpful suggestion. The figure has been updated. The legend “Figure 7I” has been supplemented.

Point 11: -Page 12, Figure 8, “Screening for shared metabolites among gut microbiota, serum, and follicular fluid”: the captions are very small and difficult to read, even after being enlarged by computer.The figure needs to be improved.

Response 1: Thanks for the helpful suggestion. The figure has been updated.

Point 12: -Page 15, Figure 9, “Metabolite levels in follicular fluid related to arachidonic acid and steroid hormones”: the captions are very small and difficult to read, even after being enlarged by computer. The figure needs to be improved. Scale bars are required for Figure 9K.“figure 9A” is not indicated in the legend.

Response 1: Thanks for the helpful suggestion. The figure has been updated. Scale bar, 20μm. The legend “Figure 9A” has been supplemented.

Point 13: -Page 19, line 563. “Following the manufacturer's instructions” is not enough for a scientific article: briefly describe the method.

Response 1: Thank you very much for your advice. Following the manufacturer's instructions has been deleted due to ambiguities caused by writing habits.

Point 14: -Page 19, line 594. “5.8. Immunofluorescence Staining”. Were negative controls (without primary antibody) performed? Indicate the results obtained.

Response 1: Thanks for the helpful suggestion.  Result figures for the negative control have been additionally displayed in the supplementary figures.
